Effects of the environmental conditions and seasonality on a population survey of the Andean condor Vultur gryphus in the tropical Andes

Márquez-Alvis Sandra 1 2
http://orcid.org/0000-0002-7229-6598 Vallejos Luis Martin 3 4 5 martin.vallej@gmail.com
Paredes-Guerrero Santiago 6
Pollack-Velasquez Luis 2 4
http://orcid.org/0000-0001-7991-8807 Santos Gabriel Silva 7 8 ssantos.gabriel@gmail.com
1 CONSERVACCION , Lima, Lima , Peru
2 Departamento de Ciencias Biológicas, Facultad de Ciencias Biológicas, Universidad Nacional de Trujillo , Trujillo, La Libertad , Peru
3 Programa de Pós-Graduação em Ecologia, Instituto de Biologia, Universidade Federal do Rio de Janeiro , Rio de Janeiro, Rio de Janeiro , Brazil
4 Departamento de Ornitologia, CINBIOTYC , Piura, Piura , Peru
5 Laboratorio de ecologia de aves y ecologia comportamental, Departamento de Ecologia, Universidade do Estado do Rio de Janeiro , Rio de Janeiro, Rio de Janeiro , Brazil
6 Reserva Nacional Pampa Galeras Bárbara D’Achille, SERNANP , Ayacucho, Ayacucho , Peru
7 Instituto Nacional da Mata Atlântica , Santa Teresa, Espirito Santo , Brazil
8 Programa de Pós-Graduação em Ecologia e Evolução, Universidade do Estado do Rio de Janeiro, Rio de Janeiro , Rio de Janeiro , Brazil
Pimm Stuart
Electronic publication date: 2023 Jan 24
Publication date: 2023
Volume: 11
Electronic Location ID: e14763
Received 2022 Jul 13; Accepted 2022 Dec 27
Copyright: © 2023 Márquez-Alvis et al.
Copyright year: 2023
Copyright holder: Márquez-Alvis et al.
License: This is an open access article distributed under the terms of the Creative Commons Attribution License, which permits unrestricted use, distribution, reproduction and adaptation in any medium and for any purpose provided that it is properly attributed. For attribution, the original author(s), title, publication source (PeerJ) and either DOI or URL of the article must be cited.
License URL: https://creativecommons.org/licenses/by/4.0/

Keywords: Age rate bird, Conservation of carrion vertebrates, Conservation and planning, Population census plans, Temperature and rainfall, South America Vultures, Vulture and condor collapse, Roosting, Raptor

Funding: Coordenação de Aperfeiçoamento de Pessoal de Nível Superior—Brasil (CAPES)—Finance Code 001 CNPq 318065/2021-5 Luis Martin Vallejos Bardales was supported by Coordenação de Aperfeiçoamento de Pessoal de Nível Superior—Brasil (CAPES)—Finance Code 001. Gabriel Silva Santos was supported by CNPq (grant number 318065/2021-5). The funders had no role in study design, data collection and analysis, decision to publish, or preparation of the manuscript.

==============================
Background

Among the New World vultures, the Andean condor is considered one of the most culturally and ecologically important species. However, their populations are declining over their entire distributional range. In response, conservation strategies have been implemented in many countries to reverse the increasing extinction risk of this species. The initiatives rely on extensive population surveys to gather basic information necessary to implement policies and to intervene efficiently. Still, there is a need to standardize the surveys based on seasonality and suitable environmental conditions throughout the species distribution. Here, we provide the first assessment of how daily temperature, rainfall, and seasonality influence surveys of Andean condors on a communal roost in the central Peruvian Andes.

Methods

Using an autoregressive generalized linear model, we associated environmental variables with visual surveys of adult and young condors at three different times of the day and three times a week between June 2014 and March 2015.

Results

We found that both adults and young Andean condors showed a threefold reduction in the use of the communal roost after the beginning of the rainy season. Colder and drier days (dry season) are preferable for surveying, as we expect the total number of condors using communal roosts to reduce under rainy (rainfall = −0.53 ± 0.16) and warmer days (temperature = −0.04 ± 0.02) days. Therefore, the significant variation in the use of roosts across seasons and hours should be carefully accounted for in national surveys, at the risk of undermining the full potential of the communal roost surveys. Moreover, we also found a strong bias towards immatures (about 76%) in the adult:immature ratio and a remarkable absence of Andean condors during the wet season. These results suggest that the species might be using other unknown communal roosts hierarchically. Such results provide key information for selecting priority areas for conservation and selecting the best time to survey this species in the tropical Andes. Finally, it may open a fruitful avenue for further research on the protection of the Andean condor.

Introduction

Globally, biodiversity is disappearing at unprecedented levels, threatening multiple species, including those that are key to society’s cultural identity. The extinction of such species is predicted to lead to significant cultural impacts in those societies (Garibaldi & Turner, 2004; Platten & Henfrey, 2009; Coe & Gaoue, 2020). In response, policymakers, scientists, and practitioners, supported by public opinion, have focused on reverting the decline of these culturally important species toward selecting priority areas for conservation and recovery plans (Smith & Sutton, 2008; Walters et al., 2010; Bennett, Maloney & Possingham, 2015; He & Guo, 2021). Because of these species’ public appeal, investing in their conservation may represent an exceptional opportunity to raise funds and establish new protected areas (Smith & Sutton, 2008; Guerra et al., 2011; Bennett, Maloney & Possingham, 2015). Indeed, conservation programs of culturally important species implemented worldwide have been successful, such as the California condor (Gymnogyps californianus) in the USA (Walters et al., 2010) and the golden lion tamarin (Leontopithecus rosalia) in Brazil (Kierulff et al., 2012, but see others examples in Bowen-Jones & Entwistle, 2002; Infield et al., 2018). Similarly, the recent conservation initiatives to protect the Andean condor have great potential to succeed.

The Andean condor (Vultur gryphus) is an emblematic species in the Andean countries of South America (Ibarra et al., 2012; Jacques-Coper, Cubillos & Ibarra, 2019). The species has a wide latitudinal and altitudinal distribution with vital historical and cultural relevance (e.g., tourism, scavenging, and art; Donázar et al., 2016; Michel, Whelan & Verutes, 2020). However, Andean condor populations are decreasing (BirdLife International, 2022). Recently, the globally threatened status of the Andean condor changed from Near Threatened (NT) to Vulnerable (VU) in the IUCN Red List (BirdLife International, 2022). In addition, a more critical classification is reported in Peru and Ecuador as Endangered (EN) (Freile et al., 2012; SERFOR, 2018) and in Venezuela and Colombia as Critically Endangered (CR) (Rodríguez, Barrera & Ciri, 2005; Renjifo et al., 2016). Large specialized vultures, such as the Andean condor, are the most threatened functional guild among birds in the world (Buechley & Şekercioğlu, 2016). Poisoning and lead contamination are the main threats to the Andean condor (Buechley & Şekercioğlu, 2016; Plaza & Lambertucci, 2020). To reverse the current threat status of the Andean condor, several Andean countries (e.g., Colombia, Ecuador, Peru, Bolivia) are preparing national conservation plans and selecting priority areas for conservation (e.g., Rodríguez, Barrera & Ciri, 2005; SERFOR, 2015; Vargas et al., 2018b; Beltran-Saavedra et al., 2020). These national conservation plans begin with extensive surveys of well-known communal roosts and potential new ones. Thus, optimizing the Andean condor survey is timely to overcome financial, staff, and logistical limitations.

A standardized survey protocol is essential for fostering robust and comparable estimates of population sizes, but particularly important in large and collaborative surveys such as the National Andean condor Surveys conducted in Ecuador, Chile, and Peru (e.g., Escobar, 2014; Vargas et al., 2018a). The Andean condor’s habit of assembling in communal roosts (see “studied species and data collection” in Materials and Methods section), where most of these surveys were conducted, provides a low-cost and non-invasive opportunity for gathering data for science-based conservation plans (Bibby et al., 2000, but see Perrig et al., 2019). However, there are no specific protocols for these surveys (but see Wallace et al., 2022), which could lead to erroneous conclusions about what communal roosts we should prioritize for conservation (Miguel et al., 2019; Wallace et al., 2020). In addition, scarce financial resources, logistic requirements, and trained staff limit long-term surveys (Ellis & Taylor, 2017; Vieira, 2020). Long-term surveys can be more cost-effective if we identify appropriate surveying windows, which requires some knowledge of the species’ behavior (e.g., Santos et al., 2009; Ellis & Taylor, 2017; Mazumdar, Ghose & Saha, 2017). It includes understanding how climate variables (e.g., temperature and rainfall) affect the roosting behavior in the case of the Andean condor. By considering these factors, surveyors can better plan their observations and collect more accurate data.

Like other vultures, Andean condors rely on thermals during their gliding to keep and conserve energy for different behaviors. Thus, external factors, such as environmental variables (e.g., season, temperature, and rainfall), are essential for the species (e.g., Wallace & Temple, 1988; Lambertucci, Ruggiero & Nascimento, 2013; García-Jiménez, Pérez-García & Margalida, 2018). Particularly for the Andean condor, the temperature is related to the time spent foraging (Shepard et al., 2011) and flying (Perrig et al., 2020). In addition, several studies in subtropical regions have reported seasonal differences in the use of communal roosts (Sarno, Franklin & Prexl, 2000; Kusch, 2004; Lambertucci, Jácome & Trejo, 2008; Lambertucci, 2010; Pavez, 2020). Thus, we expect climatic variables to affect the effectiveness of Andean condor surveys (i.e., the proportion of birds encountered during the surveys). Still, there is no information assessing this so far, particularly in the tropical Andes, where studies on vultures are scarce (Santangeli et al., 2022). As a result, surveys conducted under unfavorable climatic conditions or in less desirable seasons may underestimate the conservation importance of communal roosts (see Rogers et al., 2006; Villén-Pérez et al., 2013; Wood et al., 2017).

Here, we used count data from Andean condor surveys and an Autoregressive Generalized Linear Model to investigate the effects of season, temperature, and rainfall on the number of Andean condors observed in a communal roost in the tropical Andes of Peru. As mentioned above, these variables affect the behavior of the Andean condor, but it is not known how they impact surveys of this species. Therefore, our findings will allow us to determine the time, climatic conditions, and seasons that are optimal for conducting surveys in order to gather accurate data for the national surveys. Furthermore, because we are studying a poorly sampled area, we also provide valuable information about the importance of the studied site for the Andean condor conservation in the tropical Andes of Peru.

Materials and Methods

Studied species and data collection

The Andean condor is the largest scavenger in South America (Winkler, Billerman & Lovette, 2020). The behavior of aggregating in communal roosting is widespread among vultures and has been particularly useful for monitoring this species (Perrig et al., 2019). Environmental conditions can influence this behavior (Beauchamp, 1999; Mazumdar, Ghose & Saha, 2017), resulting in thermoregulation benefits and low energetic costs to obtain partners and available resources (i.e., food and mates; Blanco & Tella, 1999; Beauchamp, 1999; Moleón, Bautista & Madero, 2011). Particularly for the Andean condor, researchers commonly find the communal roosts in areas with high avian diversity (Lambertucci, Ruggiero & Nascimento, 2013) and have been associated with philopatric behavior (Padró et al., 2019) and age-class segregation (Lambertucci, 2013). However, most of the existing knowledge about the Andean condor’s biology is limited to studies in the subtropical Andes. At the same time, tropical areas (e.g., Peru) remain poorly studied (Santangeli et al., 2022).

The study site is located at the Pampa Galeras-Bárbara D’Achille National Reserve (RNPG-BA)—SERNANP Ayacucho and is already recognized as a priority area for the conservation of Andean condors (Wallace et al., 2020). It is located in the highland of the Andean Puna ecoregion, in the central Andes of Peru, at 4,100 m (Fig. 1). This area presents hills from 50 to 150 m and slopes from 40° to 65°. In this cold-dry area, the minimum temperature is −8 °C, and the maximum is 16 °C, with an annual rainfall of 450 mm usually limited to September to March (SERNANP, 2015). Winds can reach speeds between 2–40 km/h (SERNANP, 2015). The study site represents the unique communal roost known nearby the RNPG-BA. The other two nearest communal roosts are in the locations of “Sondondo” and “Punta Gallinazo” (Reserva Nacional San Fernando), at 60 km north-northwest and 130 km southeast from the RNPG-BA, respectively. These nearest communal roosts have distinct climatic conditions and represent, for the Andean condor, a long journey from one roost to another despite its high mobility (the mean daily flight is 64 km, with a maximum range of 197 to 350 km; Lambertucci et al., 2014; Perrig et al., 2020; Padró et al., 2018).

Figure 1 Study area.

The Andean condor roost survey and SENAMHI Meteorological Station. Pampa Galeras Barbara D’Achille National Reserve - SERNANP, Lucanas, Ayacucho, Peru. The image shows different Andean condor individuals roosting early in the morning. Map data © 2022. Imagery © 2022 Landsat/Copernicus, Data SIO, NOAA, U.S. Navy, NGA, GEBCO, Imagery © 2022 TerraMetrics. Photos by: Santiago Paredes.

The Andean condor surveys were conducted from November 2015 to March 2016, three times a week (=109 days), using binoculars (10 × 50 mm). We performed three surveys daily: 06:00 a.m., 12:00 p.m., and 06:00 p.m. (modified from Lambertucci, Jácome & Trejo, 2008). The early morning and late afternoon survey hours corresponded to the period most individuals were roosting (i.e., 06:00 a.m. and 06:00 p.m.). Also, we used the midday survey (12 p.m.), when Andean condors seldom use roosts, to compare their roost abundance with the other two time periods of the day (Sarno, Franklin & Prexl, 2000; Kusch, 2004). The roosted individuals were counted and separated into adults, with black and white plumage, and immatures, with brownish-gray plumage (see Perrig et al., 2019; Houston et al., 2020). Because distinguishing between males and females was difficult at certain distances and during the sunset, we decided to avoid distinguishing males and females in this study. We used surveys to estimate: (1) the minimum population size in the communal roost, calculated as the maximum number of adults added to the maximum number of immatures recorded, and (2) the immature/adult ratio.

Parallel to the surveys, we obtained the temperature and rainfall data from the Servicio Nacional de Meteorologia e Hidrologia (SENAMHI) weather station. This station is located in the facilities of the RNPG-BA checkpoint, 7 km southwest of the communal roost studied. Climatic data were available at 7:00 a.m. and are provided in the Supplemental Material.

Statistical analyses

We used a two-way ANOVA to compare the number of Andean condors surveyed across seasons and time of the day. Fitted values were plotted against the residuals to check for unequal error variances and outliers (see Supplemental Materials S1, S2). In addition, we assessed the extent to which temperature, rainfall, and seasonality predict the number of condors in the communal roosts using generalized linear models. Because the Andean condor survey resulted in a time series, we compared competing autoregressive generalized linear models (Autoregressive GLMs hereafter). Since we had no prior hypotheses about how climatic variables and seasonality would affect surveyed condors, we built Autoregressive GLMs representing all possible combinations of effects on the climatic variables to the Andean condor survey. We compared these competing models using the small sample corrected Akaike Information Criterion (AICc). We ranked the models according to the differences (∆AICc) of the model with the lowest AICc, which was assumed to be the most plausible. We considered models with ∆AICc < 2 as equally plausible. The Akaike’s weight (w) shows the weight of evidence for a particular model. Procedures to classify the dry and wet seasons and the rationale of the Autoregressive GLM are better explained next. All analyses were conducted using software R 4.1.0 (R Core Team, 2021).

We evaluated the influence of environmental variables on condor surveys using an Autoregressive GLM with a negative binomial distribution from the tscount package to model the total condors surveyed (version 1.4.3) (Liboschik, Fokianos & Fried, 2017). It is preferable to use autoregressive models when dealing with time series, such as the Andean condors surveys. A time series implies that the number of individuals y observed at time t, (yt) is connected to the number of individuals observed in previous i observations (yt-i), where i is the time lag included in the autoregressive model. The first step to modeling an autoregressive GLM is defining the number of time lags in the model, informed by a preliminary partial autocorrelation function (PACF). Although the number of time-lags parameters of the model in the autoregressive GLM must be prior informed, the model will estimate the magnitude to which each time-lag determines the observed values in yt. The number of time lags included in the model is necessary to remove the temporal autocorrelation between the residuals. Therefore, we evaluated the partial autocorrelation using the forecast package (Hyndman & Khandakar, 2008; Hyndman et al., 2022), which suggests that observation yt-1 is strongly related with one to four observations back in time i, i = 1, 2, or 4 (see Supplemental Material S1). Using i = 1 in the models as our first attempt was sufficient to remove the autocorrelation between the residuals (see Fig. S1.7), so we did not make other attempts. Because our first attempts using Poisson distributed models showed overdispersion, we built all models with a negative binomial distribution for a better fit (see Fig. S1.8).

Visual inspection of the number of surveyed condors unveils a gradual decrease over the year from about October, coinciding with an increase in the rainfall and an increase in the temperature, which no longer achieves negative values (Fig. S1.1). Thus, to define a breakpoint between the dry and wet seasons, we have used the function cpt.meanvar as default from the changepoint package (Killick & Eckley, 2014). We used visual inspection and a change-point analysis to define the dry and wet seasons. Using cpt.meanvar, we detected shifts in the number of condors, temperature, and rainfall based on changes between mean and variance. This approach allows us to establish an arbitrary breakpoint between the dry and wet seasons with statistical support. The breakpoints detected by the changepoint package were on 22 October for surveyed condors, on 27 August for temperature, and on 29 August for rainfall. Since we expected multiple factors influencing the use of the communal roost over the year by the Andean condors, including some not assessed in this study (i.e., food availability and mating), we assumed we could not define a breakpoint between the seasons strictly based on climatic variables. Then, we used the mean value of the three breakpoints (i.e., temperature, rainfall, and surveyed condors) to determine the end of the dry season and the beginning of the wet season, represented by mid-September (15 September). Therefore, the dry season in our study ranged from June to mid-September and the wet season ranged from mid-September to March. Despite the use of a broader sense of classification regarding the seasons, our delimitation is supported by other studies in the tropical Andes (e.g., wet season: Oct-Mar. (Jomelli et al., 2012); and Sep-Apr, (Silva, Takahashi & Chávez, 2008)).

Results

Surveys at 6 a.m. (surveyed condors, N = 7.13 ± 7.49; mean and standard deviation) and 6 p.m. (N = 7.59 ± 7.59) significantly recorded higher numbers of condors compared to surveys carried out at 12 p.m. (N = 2.12 ± 4.34) (ANOVA, FSurvey = 168.0, df = 2, p < 0.01, Fig. 2). The slight difference between the total surveyed condors at 6 a.m. and 6 p.m. did not differ significatively (Supplemental Material S1, Fig. S1.3). Moreover, we only detected a marginally significant difference in the proportion of adults along the day (ANOVA, FSurvey = 2.51, df = 2, p = 0.08) due to differences in surveys at 12 p.m. compared to other surveys (see Figs. 2 and S1.3). Therefore, we performed our analyses using the surveyed condors at 6 a.m., although the Supplemental Materials also include comparative results for 6 a.m. and 6 p.m. surveys (Supplemental Material S2).

Figure 2 Survey time and season affects the Andean condor Census.

A comparison of adults, immatures, and total surveyed condors as well as the proportion of adults during dry and wet seasons in multiple survey times (6:00 a.m, 12:00 p.m, and 6:00 p.m). Mean values and confidence intervals at 95% are presented. Detailed statistical analyses are presented in Supplemental Material S1 (Table S1.3, Fig. S1.3).

The minimum population sizes estimated for the communal roost were 38 individuals (10 adults and 28 immatures) for the dry season and 35 (eight adults and 27 immatures) for the wet season. Estimates suggest an adult:immature ratio of 1:2.80 in the dry season and 1:3.37 in the wet season. There was a three-fold difference in the mean number of individuals recorded during the wet (N = 4.01 ± 5.52, ANOVA, FSeason = 168.00, df = 1, p < 0.01) and dry seasons (N = 14.00 ± 6.69). The proportion of adults surveyed in the dry season was 0.174 ± 0.11 and almost doubled in the wet season: 0.31 ± 0.27 (ANOVA, FSeason = 9.92, df = 1, p < 0.01). The major reason for this change was the dramatic difference in the number of immature individuals surveyed throughout the year: five-fold higher in the dry season (N = 11.70 ± 5.68) than in the wet season (N = 2.81 ± 4.23; see Fig. 2). On the other hand, adult individuals reduced much less than immature ones (from N = 2.32 ± 1.51 in the dry season to N = 1.20 ± 1.72 in the wet season; see Fig. 2). Regardless of fluctuation, results suggest a clear bias to immatures over adults along the year.

The best autoregressive GLM model supports that temperature and rainfall play an important role in the total surveyed condors in the study site (see Table 1, Fig. 3). The best autoregressive GLM supports a strong negative contribution of rainfall to the total number of condors surveyed (effect size, βRainfally = −0.53 ± 0.16, t = 3.50, p < 0.01; calculated by normal approximation as suggested by Altman & Bland (2011)). Also, our results suggest a marginal contribution of temperature to the Andean condor survey (βTemperature = −0.04 ± 0.02, t = 3.96, p-value = 0.10). Two models were considered equally plausible (∆AICc < 2): the model with the lowest AICc, considering rainfall and temperature, and another one considering rainfall and seasonality (Table 1). Moreover, different combinations of temperature, rainfall, and season consistently appear more plausible than the null model, highlighting the importance of these variables. In general, the simplest models performed better.

Table 1 Competing autoregressive generalized linear models ranked by level of support (AICc).

Comparison of the Autoregressive Generalized Linear Models used to predict the total Andean condor surveyed throughout the year. Each model is ranked according to the support provided by the small sample corrected Akaike Information Criterion (AICc) and displayed together with its degree of freedom (df), weights, and cumulative weight (Cum. weight). The first two models present the best fit for our data (∆AICc < 2).

Model	df	AICc	ΔAICc	Weight	Cum. weight	Log-Like.	
Temperature + Rainfall	4	532.08	0	0.44	0.44	−260.70	
Rainfall + Season	4	533.92	1.84	0.18	0.62	−261.62	
Temperature + Rainfall + Season	5	534.09	2.01	0.16	0.78	−260.57	
Rainfall	3	535.65	3.57	0.07	0.85	−263.60	
Rainfall + Season × Rainfall + Season	5	535.99	3.91	0.06	0.91	−261.52	
Temperature + Rainfall + Rainfall × Temperature	5	536.65	4.56	0.04	0.95	−261.85	
Temperature + Rainfall + Rainfall × Temperature + Season	6	538.76	6.68	0.02	0.97	−261.74	
Temperature + Rainfall + Rainfall × Temperature + Season × Temperature + Season	7	539.34	7.26	0.01	0.98	−260.84	
Temperature + Rainfall + Rainfall × Temperature + Season × Rainfall + Season	7	540.45	8.37	0.01	0.99	−261.39	
Temperature + Rainfall + Rainfall × Temperature + Season × Temperature + Season × Rainfall + Season	8	541.16	9.08	0.00	1.00	−260.52	
Temperature	3	544.87	12.79	0.00	1.00	−268.21	
Temperature + Season	4	546.36	14.28	0.00	1.00	−267.84	
Season	3	546.41	14.33	0.00	1.00	−268.98	
Temperature + Season × Temperature + Season	5	547.4	15.31	0.00	1.00	−267.22	
NULL	2	548.1	16.02	0.00	1.00	−270.92	

Figure 3 Time serie plot of surveyed and estimated condor.

Time-series plot of predicted and observed Andean condors throughout the year. The red line represents the Andean condors predicted by an autoregressive GLM considering temperature and rainfall and is compared with the observed condors in the surveys (in black). A bootstrap with 1,000 randomizations is also presented (in grey) and shows a general adjustment of the predicted values. Photo by Martin Vallejos.

Discussion

We found a negative relationship between the number of Andean condors in the communal roost and climatic variables such as temperature and rainfall. Additionally, communal roosts are more likely to be used in the dry than in the wet season. Our results are similar to other studies of open-habitat birds and raptors (e.g., Figueira et al., 2006; Hoodless, Inglis & Baines, 2006; Calladine, du Feu & du Feu, 2012). However, they provide essential elements to guide and design better surveys and protected areas for Andean condor conservation (Wallace et al., 2022). Below we outline our results’ contribution to the Andean condor’s biology, particularly focusing on the tropical Andes.

Updates about the biology of the Andean condor

This study was the first to describe a remarked seasonality for the Andean condors using communal roost in the tropical Andes. So far, the only roost survey available for the tropics (on the arid coast of Peru, 60 km distant from our study site) showed regular use of the communal roost throughout the year (Vásquez, 2015). The difference between our study site and the communal roosts studied by Vásquez (2015) might be due to the low rainfall on the arid coast of Peru (approx. 8–80 mm/year; Olivares, Taype & Castro, 1994). Thus, in terms of seasonality, our results are more similar to other studies in subtropical areas of the Andean condor occurrence, pointing out a higher number of individuals in the wet season (i.e., Sarno, Franklin & Prexl, 2000; Pavez, 2020). This pattern, however, seems not to be unique to the Andean condor distribution (e.g., Kusch, 2004; Lambertucci, Jácome & Trejo, 2008). Therefore, it is likely that Andean condor populations in tropical areas present distinct displacement behaviors throughout the year on the coast and in the Andes. Despite the limited insight offered by our results on this matter, given that we only sampled one communal roost, Pennycuick & Scholey (1984) have already suggested that the predominant direction of the thermals might limit the connection between the coast and the highland Andes (but see Lambertucci et al., 2018).

The reduced surveyed condors in our study site during the wet season may reflect constraints imposed by the rainfall. While the Andean condors are likely to fly better and disperse more in the summer due to the thermals (De Martino et al., 2011), our study site may be temporarily unattractive due to the higher incidence of rainfall. An alternative explanation is the reduction of individuals due to higher dispersal during the breeding time (e.g., Lambertucci, Jácome & Trejo, 2008). Despite limited knowledge about the breeding season of the Andean condor in the tropics, there is evidence of egg-laying and incubation occurring between February and June (Wallace & Temple, 1988; Ríos-Uzeda & Wallace, 2007; Restrepo-Cardona et al., 2018). Nevertheless, the absence of immature condors in the study site is the main reason for the difference in Andean condors surveyed during dry and wet seasons. This pattern suggests that breeding time alone cannot explain the differences in the number of surveyed condors we observed over time.

The adult:immature ratio found at our study site is also remarkable and sheds light on important insights about the biology and conservation of the Andean condor. The minimum population size estimated suggests an adult:immature ratio of 1:3.37, which is highly disparate compared to previous studies that have reported a ratio close to 1:1 (range, 1:0.52–1:1.50; i.e., Sarno, Franklin & Prexl, 2000; Ríos-Uzeda & Wallace, 2007; Lambertucci, 2010; Escobar et al., 2015; Pavez, 2020). For long-lived and slow-reproductive species, the adult:immature ratio usually tends to be biased towards adults, which have small annual clutch sizes (e.g., Tella et al., 2013; Margalida et al., 2020).

Because the study site represents what we consider a relatively isolated communal roost (see methods), we identified at least two potential non-competing hypotheses to explain this adult:immature bias. First, biases towards a high number of immatures may suggest good recruitment in recent years. If this is the case, our results might suggest that the population of Andean condors in RNPG-BA is increasing. Indeed, more recent observations on this site have reported up to 45 individuals (S. Paredes-Guerrero, 2022 unpublished data). Alternatively, the social hierarchy is also a potential explanation. It is well-known that adults at the top of the social hierarchy have access to the best resources, leaving lower-quality resources for the individuals at the bottom of the social hierarchy, usually the youngest ones (Kirk & Houston, 1995; Donázar et al., 1999). The communal roost conservation priority must therefore consider a population spatially separated with different communal roosts providing shelter primarily for different phases of the Andean condor life cycle if the social hierarchy is that important. Finally, we recognize both hypotheses as possible because we investigated only one communal roost; nonetheless, this must be considered in the national census in Peru or other Andean countries to clarify these pressing questions further. Such results also chorus with other studies highlighting potential healthy populations based on the adult:immature ratio in Bolivia (Ríos-Uzeda & Wallace, 2007) and Ecuador (Naveda-Rodríguez et al., 2016).

Guidance for national surveys and conservation of the Andean condor

Our results support multiple concomitant surveys of potential communal roosts at each season. These surveys should be conducted mainly in the morning or late evening to avoid underestimation; different results are found in subtropical areas (Kusch, 2004). Surveys should also consider daily climatic conditions (e.g., Perrig et al., 2020). Particularly for the study site and potentially for other sites in the tropical Andes with marked climatic seasonality, surveys on colder and dryer days are more prone to capture the full potential (i.e., more individuals) of the Andean condor’s communal roosts. On the other hand, seasonality might not be a concern for areas with more stable climatic conditions. Finally, as expected, given that Andean condors can travel long distances (see Piana & Vargas, 2018; Perrig et al., 2020), protected areas could be an essential conservation tool for this species, but not alone (see Guido et al., 2020; Wallace et al., 2020). Instead, a network of protected areas, including communal roosts with different environmental characteristics, is important to provide adequate protection for Andean condors (Wallace et al., 2020). In the subtropical areas, significant progress has been made, particularly in the last decade, in mapping priority areas for the Andean condor (e.g., Perrig et al., 2019) and unveiling the spatial genetic structure of the population among multiple communal roosts (Padró et al., 2018, 2019). These topics, however, remain scarcely explored in the tropical Andes, which might be particularly important to clarify the extent to which the differential use of communal roosts along the year may reflect on priority areas selection and genetic structure of the condor populations. In brief, these areas might require particular strategies to promote an efficient national survey.

Conclusions

We studied the effect of daily environmental conditions and seasonality on the Andean condor surveys in a communal roost in the tropical Andes. We found that temperature, rainfall, and seasonality affect the Andean condors surveyed. Thus, these variables must be considered to correctly assess the importance of the communal roosts during the Andean condor surveys. The highly disparate proportion of immatures compared to adults, and the dramatic reduction of the total condors surveyed during the wet season, indicate that this population uses different communal roosts during the year. The effect of environmental conditions, seasonality, and potential spatial segregation, represents important characteristics to define efficient conservation areas and surveys protocol. Due to the latitudinal and altitudinal gradient within the Andean condor distribution range, our conservation results must be extrapolated with caution.

Supplemental Information

Supplemental Information 1 Supplemental figures, tables, script and the raw data.

The raw data collected in the Pampa Galeras-Bárbara D’Achille National Reserve includes the climatic variables from the Servicio Nacional de Meteorologia e Hidrologia (SENAMHI) and is described in detail in the Medata.pdf file

Click here for additional data file.

We thank Servicio de Áreas Naturales Protegidas Por El Estado [SERNANP] for providing the Andean condor survey data and the official park rangers and volunteers who assisted in data collection, especially Mr. Hernán Sosaya. We also thank Irwing Saldaña for suggestions on the methods and Thomas Defler and Danielle Moreira for their help with the English revision. Finally, we thank to Andreas Altwegg and two anonymous reviewers for their helpful comments.

Additional Information and Declarations

Competing Interests

Author Contributions

Animal Ethics

Data Availability

The authors declare that they have no competing interests.

Sandra Márquez-Alvis conceived and designed the experiments, performed the experiments, prepared figures and/or tables, authored or reviewed drafts of the article, and approved the final draft.

Luis Martin Vallejos conceived and designed the experiments, analyzed the data, prepared figures and/or tables, authored or reviewed drafts of the article, and approved the final draft.

Santiago Paredes-Guerrero performed the experiments, authored or reviewed drafts of the article, and approved the final draft.

Luis Pollack-Velasquez conceived and designed the experiments, authored or reviewed drafts of the article, and approved the final draft.

Gabriel Silva Santos conveived and designed the experiments analyzed the data, prepared figures and/or tables, authored or reviewed drafts of the article, and approved the final draft.

The following information was supplied relating to ethical approvals (i.e., approving body and any reference numbers):

Servicio nacional de areas protegidas por el estado—SERNANP.

The following information was supplied regarding data availability:

The data and code is available at Zenodo: Santos, Gabriel Silva. (2022). Effects of the environmental conditions and seasonality on a population survey of the Andean condor Vultur gryphus in the tropical Andes [Data set]. In PeerJ (1.0). Zenodo.https://doi.org/10.5281/zenodo.7446654.

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
