# Peer review of "Effects of the environmental conditions and seasonality on a population survey of the Andean condor Vultur gryphus in the tropical Andes"

_PeerJ, doi:10.7717/peerj.14763_

## Round 0.1 · original submission · Major Revisions

As you can see, neither of the two reviewers are happy with your paper. The comments of the one who recommends rejection are very detailed, however, and I think you should be able to address them. Some of them are pedantic.

Both reviewers make substantial comments and you will need to convince me that you have addressed all of them before I will consider sending your revision to reviewers a second time.

Reviewer 1 ·

Basic reporting

Authors
This papers evaluate the use of one communal roost by condors throughout the year and relate the condor’s abundance to weather conditions. The information is valuable although limited considering it is only one communal roost, so inferences are limited. However, the information provided is interesting considering that this is a rare and threatened species and there is no much information for Peru. The paper needs to be better framed in the introduction, improve the English, clarify the analyses, and improve the discussion. Another important point is that you check the references since some citations do not say what you are trying to support. I have included some suggestions regarding this in the attached document.

My main suggestions are that you start with the bigger problem/question you will face. Now the paper is written for specific ornithologists interested in condor communal roosting, so it looks more suitable for an ornithology journal. For wider journals and audience is better to frame the article in the problem and not so specifically to the study species. You can contribute to the question of why and how communal roosts are used by birds in general. You should use more general references before you get into the study species. The work by Guy Beauchamp on communal roosts and group size could be a good start (https://scholar.google.com/citations?user=kTHUyuQAAAAJ&hl=es&oi=ao). There are several hypotheses there to discuss. Apart from this, you should improve the references on condors, a good review of papers will help.

Throughout the text, there are several sentences that are not clear enough, particularly in methods and results. You should improve this, and some help from a colleague with good skills in writing in English could be helpful.

Explain better condor ecology, do they breed in communal roosts? do they have colonies or just roosts? If the breed there, how can this affect their use of the roost? If they don´t why is this important for evaluating the communal roosting behaviour?

Go back in the discussion to the general ideas, don’t focus only on your specific results obtained in one condor roost, but more general. You can use your frame from the introduction for this, and again the work by Beauchamp and others in this sense. Similarly, in the conclusion try to go further using your observations in a more general way, considering the theory of communal roosting in birds.

You will find several comments and recommendations in the attached pdf.

Experimental design

It is original but knowledge gaps are not well identified. Methods need more clarifications.

Validity of the findings

Themain concern is that it is only one roost.
Conclsuions needs to be better supported and a not only focused on the study species.

Annotated reviews are not available for download in order to protect the identity of reviewers who chose to remain anonymous.

Reviewer 2 ·

Basic reporting

- The introduction section starts by introducing the species without providing sufficient background about the general problem. This section needs to be revised/restructured.

- The authors used census throughout the manuscript including the title, but that could be misleading. Strictly speaking, census refers to complete count, which is almost impossible in wildlife. Change "census" to "count" or "survey".

- Tables and Figures captions are not self-explanatory.

Experimental design

- L105-111. The objectives of this study were not well defined. The authors need to clearly state their hypotheses and predictions.

- The authors need to improve the presentation of their statistical analysis. The current presentation lacks flow (i.e., no logical structure). The authors should present the variables followed by the description of the underlying model and model selection procedure.

- L143. It is unclear whether the authors checked the assumptions of ANOVA. Please clarify.

- L146. The authors need to better explain why the autoregressive model was appropriate for their analysis.

- L147. What do you mean by “all possible combinations”? Looking at Table 2, you did not fit any models with interaction effects.

- L149-150. This is a bit strange. AIC does not tell the models are significantly different. Please revise.

- L152-155. Please explain why a negative binomial distribution was appropriate for your data. Have you tried other probability distribution? Have you compared the results?

Validity of the findings

- L184-185. Why did not use a formal statistical procedure to test if the counts differ between different time periods?

- L189. Please clarify whether the ANOVA assumptions were checked. Note that the validity of your inference depends on whether the assumptions are met or not.

- L199-202. The use of ANOVA for proportions is inappropriate. The authors should use an appropriate method for comparing proportions.

- L209. Please avoid using the word “significant” when you present model selection results. AIC does not tell the difference between two models are significant. Please check other published papers how to report model selection results. Also, please revise Table 2 caption.

- Table 2. The number of parameters (K) looks incorrect for some of the models. Could you please explain why K is 2 for the null model? Please double check K for each model.

- Table 2 caption. Here you mentioned a Poisson distribution, but in the methods section you mentioned that a negative binomial distribution (See L153) was used. This is very confusing and needs clarification.

Additional comments

- L60. Change “increased” to “changed”.

- L131-135. It is uncommon to use bullet points in a scientific paper.

- L152. “To assess the role played by environmental variables…” This is a bit weird. Please revise.

- L158-159. What do you mean by “two observations are performed”?

- L166. “Dry and wet seasons were classified statistically…”. This is unclear to me.

- L170. Is “cpt.meanvar” a function in R?

- L175. Make “changepoint” italic. Be consistent.

- L179. ”The limited settled…”. This is unclear to me.

- L198. Does the average refer the mean or median? Please clarify.

- L199. Add “than” before “the wet season”.

---

## Round 0.2 · Minor Revisions

Please make these changes — I believe they are all minor and I hope you can do them quickly. I then expect to accept your paper.

·

Basic reporting

1. I found the language relatively clear but found a few unclear passages / language problems that I detail in the "additional comments" section.
2. I'm not familiar with the literature on condors and can't comment on the appropriateness of most references. Program R should be cited.
3. The article structure is appropriate; the figures are informative and visually appealing. The raw data are shared as an xlsx file. I recommend sharing them in a non-proprietary format, e.g. as csv files.
4. This is not hypothesis-driven work. The goal is a bit unclear but I understand that the authors want to know how the number of condors at this particular roost varies over the time of the day and season. I understand that this is meant to guide a monitoring program, presumably indicating in which season and at what time the birds should be counted to make the survey most effective (presumably find the highest proportion of birds at the roost). This could be stated more clearly. The title could also reflect this better. E.g. "When are condors present at their roost in the D'Achille National Reserve?"

Experimental design

1. The research question and goals could be stated more clearly. You mention the need for a range-wide monitoring program. Is this roost under consideration as one of the main monitoring sites? From the title, it is not clear whether it is the effect of environmental conditions on the surveys or on the birds that is under study. It seems to be on the effectiveness of the surveys. I don't know much about condor biology. The fact that the number of birds at the roost fluctuates presumably means that the birds can also roost in other places. It would then seem important to know what proportion of the local population uses this roost. Is it 5% or 95%? The data don't seem to give any information on this. It is not quite clear how the results inform the monitoring / conservation needs mentioned in the introduction.
2. The methods appear to be sound and there are no ethical considerations, as far as I can tell.
3. The methods are described in sufficient detail.

Validity of the findings

1. With the goal relatively vague, it is hard to say whether the study achieves its goal. The benefit to the literature could be more clearly stated.

2. The data and R code are provided. I found the R code neat and fairly easy to read.

Additional comments

Line 71: what is “cultural disruption”?

Line 82: what does “It is potentially the case...” refer to? Clarify.

Line 117: “the effects of climatic conditions on the species” can you be more specific here? Are you thinking about the habitat use or roosting behaviour, perhaps?

Line 128: “we expect climatic variables to affect the Andean Condor surveys” also sounds a bit vague. Are you expecting these variables to affect the effectiveness of the surveys? Or the proportion of birds encountered during surveys?

Line 178: “roosters” should be “roosts”

Line 193: “We used a two-way ANOVA to compare ... surveys across seasons and surveys” what exactly was your response variable? The counts?

Line 209: “ the influence of environmental variables in condor surveys” should be “... on condor surveys”

Line 210: you used the negative binomial for the counts adults plus immatures), presumably. And what did you use for the immature/adult ratio?

Line 214: “where yt-i is the time lag...” the time lag is just i; y is the count.

You use the word “survey” to refer to the surveys in general but also to time of the day (6am vs 12pm vs 6pm). That is confusing. Use “time of the day” or something like this for the latter.

Line 257: “surveys at 12 a.m” that would be midnight. Do you mean 12 pm, probably? You also say 12AM in the “Final data.xlsx”. Please check throughout.

Legend to Table 1: “Akaike Criteria Information” should be “Akaike Information Criterion”

You used rainfall, temperature and season as predictors. Were these variables correlated with each other?

Figure S1.1: give the units for temperature and pluviosity. Is pluviosity just rainfall or does it have a special meaning here? If the former, why not just say ‘rainfall’?

Data: the data should be provided in a non-proprietary format, e.g. as csv files to ensure that they remain readable in the future.

---

## Round 0.3 · accepted · Accept

Thank you for making your changes promptly.